# Generation of Viral Particles with Brain Cell-Specific Tropism by Pseudotyping HIV-1 with the Zika Virus E Protein

**DOI:** 10.3390/mps7010003

**Published:** 2023-12-28

**Authors:** Hai Dang Ngo, Jan Patrick Formanski, Vivien Grunwald, Birco Schwalbe, Michael Schreiber

**Affiliations:** 1Department of Virology, LG Schreiber, Bernhard Nocht Institute for Tropical Medicine, 20359 Hamburg, Germany; 2Department of Neurosurgery, Asklepios Kliniken Hamburg GmbH, Asklepios Klinik Nord, Standort Heidberg, 22417 Hamburg, Germany

**Keywords:** flavivirus, retrovirus, lentivirus, Zika virus, HIV-1, viral vector, pseudotype

## Abstract

Flaviviruses are a family of RNA viruses that includes many known pathogens, such as Zika virus (ZIKV), West Nile virus (WNV), dengue virus (DENV), and yellow fever virus (YFV). A pseudotype is an artificial virus particle created in vitro by incorporating the flavivirus envelope proteins into the structure of, for example, a retrovirus such as human immunodeficiency virus type-1 (HIV-1). They can be a useful tool in virology for understanding the biology of flaviviruses, evaluating immune responses, developing antiviral strategies but can also be used as vectors for gene transfer experiments. This protocol describes the generation of a ZIKV/HIV-1 pseudotype developed as a new tool for infecting cells derived from a highly malignant brain tumor: glioblastoma multiforme grade 4.

## 1. Introduction

Retroviruses are a family of RNA viruses. They can be genetically modified to prepare viral vectors for gene delivery or oncolysis [1]. By definition, viral vectors are virus particles that are used to introduce genetic material into target cells, usually without causing a productive infection. Pseudotyping is often used to extend the tropism of viral vectors, making them applicable for various research and therapeutic purposes. It is a common laboratory technique in which the envelope proteins of a virus are exchanged so that the resulting viral particle carries the envelope proteins of another virus [1,2]. A promising method for treating cancer diseases is the use of oncolytic viruses. Oncolytic viruses are viruses that preferentially infect and kill cancer cells. They are used experimentally as part of oncolytic cancer virotherapy to combat cancer diseases [3,4]. The aim is to infect cells in a natural way and either destroy or modify them. An important goal is the development of oncolytic viruses or viral vectors with a high specificity, a particular tropism, for the respective cancer cell.

The emergence of a serious disease, called microcephaly, has brought the Zika virus (ZIKV) into the spotlight as a new therapeutic candidate to fight brain cancer [5]. The fact that ZIKV infects specific cells in the neonatal brain that are also involved in the development of the most severe brain tumor in adults is crucial for using the ZIKV tropism to develop suitable oncolytic viruses or viral vectors [6,7,8]. Therefore, the design of ZIKV-derived envelope proteins for pseudotyping HIV-1 is an important approach. The receptor molecule AXL, a member of the TAM (Tyro3, AXL, Mer) family of receptor tyrosine kinases, plays a role in cancer progression, metastasis and treatment resistance, and high AXL expression is often linked to a poorer prognosis in brain cancer patients [9]. Most importantly, AXL is the cellular receptor that ZIKV uses to enter glioma cells [10,11].

A prerequisite for the production of HIV-1 pseudotypes is that the new envelope protein is also located at the site of HIV-1 assembly and budding just like the original gp41/gp120 envelope. However, this does not apply for the combination of HIV-1 and ZIKV. As shown in Figure 1, HIV-1 assembly and budding occurs at the cell membrane, whereas ZIKV particle formation occurs at the membrane of the endoplasmic reticulum.

For protein E, there are reports of E pseudotyped viral vectors such as JEV/MuLV or JEV/HIV [12,13], but, especially for ZIKV, there are only a limited number of publications available describing the successful ZIKV/HIV pseudotype production [8,14,15,16]. In the first publication, it was demonstrated that the ZIKV prME complex can produce pseudotypes, but at a very low level [8,15]. The general problem, that the ZIKV envelope is not normally transported to the cell surface, has been addressed with three different approaches. One approach was that E was expressed using a standard protein expression vector including an exchange of the E transmembrane (TM) domain with the TM of VSV-G [14]. The second approach was to use the TM and cytoplasmic domain of HIV-1 gp41 as a membrane anchor for the E protein [16]. The third and more simple approach was to link the E protein to the TM domain present between capsid and prM, TM(C), leaving the normal E protein stem and anchor region intact (Figure 2) [16].

The protocol describes in detail the preparation of HIVgfp pseudotypes with ZIKV E, using the pNLgfpAM plasmid (Figure A2). The use of GFP as a reporter allows easy detection of the pseudotype entry with fluorescence microscopy. The general procedure for the pseudotype production is shown in Figure 3. 

## 2. Experimental Design

### 2.1. General Prerequisites

#### 2.1.1. Plasmid DNA

The flavivirus envelope was expressed in the pME plasmid, a derivative of pcDNA3.1, controlled by the CMV promotor, and GCCGCCGCCATGG used as a Kozak Sequence (Figure A1) [8,17]. Plasmid DNA was purified from the E. coli strain DH5α, grown in 2x yt medium (5 g NaCl, 16 g Trypton, 10 g Yeast extract) using a commercial plasmid purification kit (see Section 2.4. point 5). DNA purification was performed according to the manufacturer’s protocol. Using the elution buffer from the kit (5 mL), the DNA must be eluted directly into isopropanol (3.5 mL) placed in 15 mL centrifuge tubes, mixed completely, and centrifuged for 30 min at 8 °C at 15,000× g. After centrifugation, carefully transfer the liquid to a second 15 mL tube and place the first tube upside down on a paper towel. It is important that the DNA pellet does not slip out of the tip of the tube. Use a tweezer and a small piece of sterile filter paper to carefully remove any liquid residue from the tube. The DNA pellet should now be dissolved stepwise in a total volume of 1000 µL. This is an important step because the DNA pellet can be easily lost. Add 500 µL of the Plasmid solution to a 1.5 mL reaction tube and mix with 1000 µL of 99.8% ice-cold ethanol (Carl Roth, #1HPH.1). Carefully invert the tube and observe the precipitating plasmid DNA, which will become visible as a light gray floc. The tubes should be incubated for 10 min on ice without vortexing and then centrifuged for 15 min at full speed (13,000× rpm) at RT in a microcentrifuge. Air dry the plasmid pellet by placing the tubes upside down on a paper towel. Remaining liquid can be removed with a small strip of sterile paper. Add 500 µL of H_2_O and incubate the tubes for 30 min at 70 °C and 10 min at 90 °C with open lids to remove excess ethanol and isopropanol. Use extensive vortexing and pipetting to dissolve the plasmid DNA. From the Plasmid stock solution, mix 5 µL with 95 µL of H_2_O and transfer the plasmid solution into a supermicro black cell to measure the OD at 260 nm. Dilute the original plasmid solution with H_2_O until a 1:20 dilution (5 µL plasmid stock solution + 95 µL H_2_O) gives a value of 1.0 in the OD_260_ measurements. Then the plasmid solution has a concentration of 1 mg/mL. This is an important step for preparing a correct 1 mg/mL stock solution, as high DNA concentrations can lead to erroneous OD_260_ values if the DNA is not completely dissolved in water (see note in Section 6.1). Therefore, the classical method using a UV/VIS photometer and quartz microcuvettes is recommended for the determination of plasmid concentrations. Plasmid DNA should also be tested in a standard agarose gel electrophoresis to analyze the physical integrity of the DNA. Finally, store the plasmid DNA in aliquots each heated at 90 °C for 10 min prior to transfection of COS-1 cells.

#### 2.1.2. Cells for Viral Packaging

For pseudotype production, we used COS-1 cells because we found that these cells were more robust during the transfection process compared with HEK293T. Another reason is an effect we have observed in regard to HIV nef. In previous studies, we compared the efficiency of pseudotype production using HIV nef to that using other pNL4-3 plasmids (i.e., pNL4-3.Luc.R-E-). We observed that the presence of HIV-1 nef was necessary to get reproduceable pseudotype transductions. This could not be achieved by using plasmids with a nef-negative phenotype. The nef-dependent effect was more significant in HEK cells, which is a second reason to use COS-1 instead of HEK293T. The pNLgfpAM plasmid is expressing gag, gagpol and the accessory proteins rev, tat, vpu, vpr, vif and nef.

#### 2.1.3. Cells for Transduction

Research into a possible link between ZIKV and glioblastoma cells has attracted considerable interest because ZIKV injections were shown to reduce the size of cancerous tumors in mice, while leaving normal, healthy brain cells unaffected [5]. This was an exciting finding because the virus seemed to have an innate preference for these cancer cells. Since glioblastoma is a very aggressive brain tumor for which there is no treatment option, every new option should be used to combat this tumor. We have established a 2D primary cell culture model to study pseudotype entry of freshly isolated GBM cells. Therefore, the GBM tissue samples were minced and the cell mixture was incubated in hCSF/DMEM 5% FBS medium. The primary cell culture, AKH-16, used in this manuscript was tested positive for Axl, Integrin αvβ5, Nanog, Nestin, Oct4 and Sox2 [15], typical markers proposed as ZIKV receptors and markers for GBM tumor cells [18].

### 2.2. Equipment

Heraeus microcentrifuge, Biofuge pico #11332 (Thermo Scientific, Dreieich, Germany).Biological Safety Cabinet, Sterilgard III Advance Class II (Baker Company, Utrecht, The Netherlands).Fluorescence microscope, EVOS M7000 (Thermo Fisher Scientific, Dreieich, Germany).CO_2_ Incubator CB 150 (BINDER, Tuttlingen, Germany).UV-VIS Spectrophotometer, ShimadzuUV160A, with supermicro cell holder for measurement of 100 µL samples. Using a supermicro black cell of fused silica, 10 mm path, #206-14334. (Shimadzu, Duisburg, Germany).Micropipettes, Eppendorf Research^®^ plus, 0.5-10 µL, 5-100 µL (Eppendorf, Hamburg, Germany).

### 2.3. Materials

Pipette tips, filter tip, transparent, Biosphere^®^ plus, low retention, 10 µL, 100 µL, 200 µL (Sarstedt, Nümbrecht, Germany).Reaction tubes, SafeSeal reaction tube, 1.5 mL, PP #72.706Sarstedt AG & Co. KG (Sarstedt, Nümbrecht, Germany).Cell culture plates, 24-well, standard surface, flat base #83.3922 (Sarstedt, Nümbrecht, Germany).Cell culture plates, 96-well, Cell+ surface, flat base, #83.3924.300 (Sarstedt, Nümbrecht, Germany).Cell culture pipettes, 5 mL, 10 mL, 25 mL (Sarstedt, Nümbrecht, Germany).

### 2.4. Reagents

Cell culture medium, DMEM, w: 4.5 g/L glucose, w: stable glutamine, w: sodium pyruvate, w: 3.7 g/L NaHCO3, #P04-4515, (Pan-Biotech, Aidenbach, Germany).Medium supplement, FBS Good, EU-approved regions, filtrated bovine serum, 0.2 µm sterile filtered PA0-37500 (Pan-Biotech, Aidenbach, Germany).Human cerebrospinal fluid (hCSF) (Asklepios Heidberg Nord, Hamburg, Germany). Use of hCSF was approved by the Ethical Commission of the Hamburg Medical Chamber (Hamburg, Germany) with registration number PV6041.Transfection reagent and dilution buffer, ScreenFect^®^A (SFA) (ScreenFect, Eggenstein-Leopoldshafen, Germany).Transfection-grade plasmid DNA isolation, NucleoBond Xtra Midi kit, #740410.100 (Macherey-Nagel, Düren, Germany).

### 2.5. Plasmids

pNLgfpAM, Nikolas Friedrich, Alexandra Trkola, Institute of Medical Virology, University of Zürich, Switzerland.pME-Z1, LG Schreiber, Bernhard Nocht Institute for Tropical Medicine, Hamburg, Germany [8].pME-E2, LG Schreiber, Bernhard Nocht Institute for Tropical Medicine, Hamburg, Germany [16].pCMV-VSV-G, addgene plasmid #8454, (addgene, Watertown, MA, USA).

### 2.6. Cells

COS-1, Leibniz Institute DSMZ-German Collection of Microorganisms and Cell Cultures, no.: ACC 63.AKH-16, primary cell culture made from tissue samples of a GBM tumor. LG Schreiber, Bernhard Nocht Institute for Tropical Medicine, Hamburg, Germany. Use of human tumor samples was approved by the Ethical Commission of the Hamburg Medical Chamber (Hamburg, Germany) with registration number PV6041.

## 3. Procedure

### 3.1. Production of Pseudotyped HIV-1 Particles

#### 3.1.1. Day One

Add 1000 µL of a COS-1 cell suspension (see note in Section 6.2) in DMEM/10% FBS to a well of a 24-well plate. The cell suspension should be adjusted so that the cells do not become too dense on day 3, as there is a risk that they will otherwise detach. If the density is too low, the transfection will result in low pseudotype production. Therefore, it is recommended to perform preliminary experiments to investigate the growth of COS-1 cells under the respective laboratory conditions to get the best possible transfection results. However, the cells should have a confluence of 70–80% on the day of transfection.

#### 3.1.2. Day Two

Replace the old medium with 1000 µL of fresh DMEM/10% FBS medium.For the transfection procedure, the DNA and SFA reagent preparations are each mixed in a separate 1.5 mL reaction tube following the recipe shown in Table 1. The plasmid DNAs are all adjusted to a concentration of 1 mg/mL in H_2_O. Because of the cell toxicity of SFA, the amount of SFA reagents should not be increased to minimize the risk of COS-1 cell detachment.Mixture 2 is added to mixture 1 in 15–20 µL increments and mixed each time by pipetting up and down (5 × 20 µL pipette strokes). Mixture 2 must be slowly transferred into mixture 1 in small steps and mixed well to prevent precipitation of the DNA. The DNA/SFA mixture will be incubated for 20 min at room temperature. After incubation, carefully add the DNA/SFA mixture onto the 1000 µL of the COS-1 cell culture medium. Do not mix the DNA/SFA mixture with the cell culture medium by pipetting up and down. Gently agitate the 24-well plates back and forth 2–3 times and transfer the plates carefully into the CO_2_ incubator. Incubate the 24-well plates for 24 h at 37 °C and 5% CO_2_.

#### 3.1.3. Day Three

The medium also containing the DNA/SFA mixture is carefully replaced with 1000 µL of fresh DMEM/10% FBS and the cells are incubated for another 48 h at 37 °C and 5% CO_2_. In this step, it is important to remove any free plasmid DNA that may be present in the cell culture supernatant. This is particularly important when using plasmids containing firefly luciferase.For the pseudotype experiments, the 96-well plates should now be prepared. Therefore, 200 µL of an AKH-16 cell suspension grown in CSF-DF medium (50% hCSF, 45% DMEM, 5% FBS) was added to a well of a 96-well plate. CSF-DF, a 1:1 mixture of DMEM/10% FBS and hCSF has been shown to be an excellent medium for the growth of cell cultures made from glioblastoma tissue samples that maintain their heterogeneity over a long period of time.

### 3.2. Transduction of GBM Tumor Cells

#### 3.2.1. Day Five

To verify successful pNLgfpAM transfection and pseudotype production, examine the wells using the EVOS M7000 fluorescence microscope (GFP channel, see note in Section 6.3).To collect pseudotype particles, carefully remove the cell culture medium from the transfected cultures without touching the bottom of the well with the pipette tip to avoid picking up of COS-1 cells. The cell culture supernatant is collected into a 1.5 mL reaction tube and centrifuged at maximum speed (13,000 rpm) for 2 min at RT. After centrifugation, carefully transfer the medium to a new reaction tube by touching the surface of the medium with the pipette tip and, finally, leaving 50–100 µL of the liquid at the bottom of the tube. Avoid picking up liquid from the bottom of the tube. The supernatant is now centrifuged for a second time (13,000 rpm, 2 min, RT) and the supernatant is again transferred into a new reaction tube. This procedure yields 800–900 µL of pseudotype containing cell culture medium for every transfected 24-well.For optimal transduction, the confluence of the primary AKH-16 cell cultures should be around 70–80% on the day the pseudotype is added. Depending on the pseudotype yield, the supernatant can be used directly or diluted in medium. As a rule, 100 µL of a pseudotype-containing medium is added to the tumor cells and the 96-well plates are incubated at 37 °C and 5% CO_2_ for 24 h.

#### 3.2.2. Day Six

The next day, the CSF-DF medium is changed, and the plates are incubated for another 48 h at 37 °C and 5% CO_2_. This step is important especially when firefly luciferase is used as a reporter gene. In this case, it is important that the cell culture supernatants are free of any luciferase activity. Therefore, several washing steps may be required 24 h after transfection to remove luciferase activity from the cell culture supernatant while leaving the cell layer intact.

#### 3.2.3. Day Eight

For the detection of pseudotype transduced cells using GFP as a reporter, the plates including the medium are examined using the EVOS M7000 fluorescence microscope using the GFP (470/525 nm) and DAPI (357/447 nm) channels (excitation/emission). The 96-well area is scanned at 4× magnification and the pictures are automatically stitched together by the EVOS M7000 imaging software. The successful transduction of AKH-16 cell cultures with Z1-HIV*gfp*, E2-HIV*gfp* or G-HIV*gfp* pseudotypes is shown in Figure 4.

## 4. Discussion

In summary, due to their versatility and safety, viral pseudotypes are valuable tools in virology and, more specifically, suitable tools in cancer research. They allow the study of oncogenic or oncolytic viruses, gene delivery, evaluation of targeted therapies, vaccine development and much more, ultimately contributing to a better understanding of cancer biology and the development of new cancer treatments. ZIKV is here used as an example that is of utmost importance in regard to brain cancer [16] but might also be of high relevance for other cancer types [14]. We have established this protocol for the development of ZIKV/HIV pseudotypes as a tool for virotherapy of brain tumors, especially glioblastoma. In this context, we were guided by the studies in which hCSF was used as a culture medium supplement [19]. In general, it is more difficult to obtain glioblastoma tumor samples on a regular basis than it is to obtain hCSF.

Unfortunately, protocols for the production of pseudotypes with flavivirus envelopes have been published in the past; their use has been unsuccessful, or data have been published questioning the production of such pseudotypes in general [20,21]. Thus, there could be several reasons for such negative results. First of all, it seems to be important to use a nef+ vector system. Nef can be expressed by the gRNA/packaging plasmid but can also be substituted by a separate plasmid. For HIV-1 based pseudotype production, the vector pNL4-3.Luc.R-E- vector [22] is often used since, together with VSV-G (pMD2.G addgene #12259), this leads to a sufficient amount of pseudotypes. But this popular vector does not express nef. It should be mentioned that the most used gag-pol packaging vector psPAX2 (addgene #12260) is also negative for nef. Thus, when used together with plasmids like pLenti-luciferase-P2A-Neo (addgene #105621), the HIV-1 accessory protein nef will always be absent. Therefore, it is now understandable that the above-mentioned protocols do not work satisfactorily.

Another aspect seems to be related to the presence of the ZIKV transmembrane domains of the prM protein. When prM is eliminated, ZIKV E alone can produce efficient amounts of pseudotypes [16]. When compared to VSV-G, the ZIKV E pseudotypes E2-HIV*gfp* produced significantly more GFP-positive foci than G-HIV*gfp*. The stem and TM domain of flavivirus E was well studied, showing that it is for the stability of the E trimer on the viral surface [23]. It is conceivable that the E-TM supports travel of the protein to the cellular membrane, whereas prM-TM could be relevant to hold the envelope complex at the ER membrane. However, this is only a hypothesis based on our experience and would need to be investigated in more detail. This hypothesis could explain the lower infectivity of the prME pseudotypes compared to VSV-G and ZIKV E. Thus, the transmembrane region of ZIKV E would not be responsible for the arrest at the ER membrane. Some reports show different results, both increased [14] and decreased activity [20], when the TM region was exchanged for that of VSV-G. In contrast, good results were obtained with a chimera of ZIKV E and the TM region of gp41 [16]. However, it has been shown that a pseudotype, E2-HIVgfp, with brain tumor specificity can be produced that has a higher transduction efficiency compared to the commonly used VSV-G pseudotypes.

Various expression plasmids can be used to render cells positive for flavivirus E. We followed the protocol published by Hu et al. [20], using a modified version of pcDNA3.1. To support transient expression in COS-1 as well as in HEK293T cells the *Pvu*II-*Pvu*II segment coding for SV40 elements was removed. All our flavivirus envelopes are expressed in this plasmid under the control of the CMV promotor. The start sequence is (GCC)_3_ATGG, which is one of the two most frequent Kozak sequences in vertebrates (GCCGCC[A/G]CCATGG) [24]. Therefore, it also seems to be important to consider the expression efficiency of E proteins when producing such pseudotypes. Therefore, it is important to develop the best possible protocol for each env/viral vector combination. The protocol presented here works with all ZIKV env constructs and the two HIV-1 vectors which we published in the paper Grunwald et al. [16]. The presented protocol has been successfully applied several times by different collaborators and the production of ZIKV E pseudotyped HIV-1 is now also part of a practical course for students. Therefore, we believe that our protocol can provide an important contribution to flavivirus research.

## 5. Conclusions

The presented protocol can be considered as a methodical breakthrough for glioma research and flavivirus research in general. Now, new studies are possible to further investigate the interaction between the viral envelope proteins and cellular receptors. In addition, the pseudotype can be exposed to antibodies or sera from individuals or animals that have been exposed to the actual flavivirus to test the efficacy of the immune response in new neutralization assays. Thus, flavivirus E pseudotyped HIV-1 particles can be used to screen potential antiviral entry inhibitors as well as vaccine candidates. Because wild-type flaviviruses are highly pathogenic and pose a safety risk, pseudotypes provide a useful virological tool while ensuring a safer laboratory environment. This allows studies to be performed in normal safety level 2 laboratories, which greatly facilitates work on flaviviruses by eliminating the need for class 3 laboratories.

## 6. Notes

### 6.1. Making Plasmid Stocks for Transfection

High plasmid concentrations are often viscous, leading to challenges in handling, transferring and measuring. Following the Machery-Nagel protocol for midi DNA isolation, the plasmid DNA eluted from the column is precipitated with isopropanol. We observed that the DNA pellet formed after centrifugation does not readily dissolve in H_2_O. Extensive pipetting and several heat incubations at 70 °C should result in a homogeneous plasmid solution. If the solution forms viscous threads during pipetting, the DNA is not completely dissolved in H_2_O. The stock solution should have an OD_260_ = 1.0 at a 1:20 dilution and an OD_260_ = 0.5 at a 1:40 dilution. If the 1:40 dilution shows an OD_260_ higher than 0.5, the plasmid DNA is not properly dissolved.

### 6.2. Making Plates for Transfection and Transduction

A method for preparation of adherent cell layers is that a known number of cells is given to microplate wells. We are not counting cells. Since only some wells are used from a 24-well microplate, we are making dilutions of COS-1 cells and are using those wells that show a 70–80% confluency on the day of transfection. Cells completely covering the surface of a 75 cm^2^ flask are solubilized in 30 mL medium. From this stock, 3 (for 8 wells) or 4 different dilutions (for 6 wells) are made. The advantage is that higher dilutions can also be used for transfections when they reach a density of 70–80% on the following days.

### 6.3. Checking Transfection Efficiency

When using GFP as a reporter, transfection can principally monitored by fluorescence microscopy as described in Section 3.2.1. However, it is not recommended to optimize the transfection protocol for the highest number of GFP+ COS-1 cells. We found that an optimized COS-1 transfection, i.e., a maximum number of GFP+ COS-1 cells or a high concentration of firefly luciferase when used as a reporter, does not directly correlate with an optimal pseudotypes yield. For production of ZIKV/HIV pseudotypes, we have observed a negative correlation [8]. Therefore, COS-1 transfection with the two plasmids should preferably be optimized on the basis of transduction efficiency using a ZIKV-permissive cell line such as U87 or a similar one.

## Figures and Tables

**Figure 1 mps-07-00003-f001:**
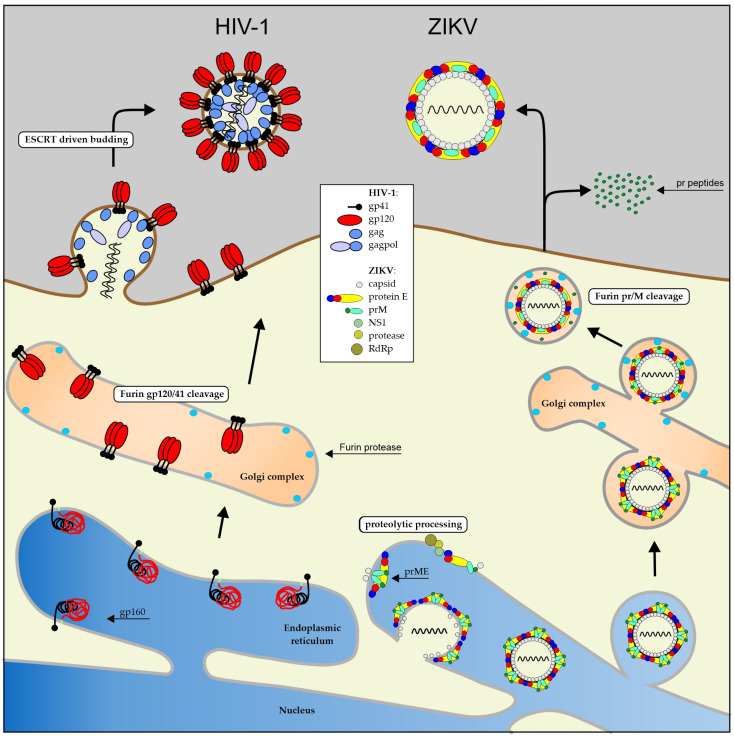
The two different ways that the viral envelopes of HIV-1 and ZIKV are formed. The HIV-1 gp160 is cleaved and modified at the ER membrane into a gp120/gp41 trimer. The gp120/41 trimer is then transported to the cellular membrane, where it binds gag and gagpol precursor proteins. The complex is squeezed out of the cell by the ESCRT machinery via the p6 domain in the gag protein. Flaviviruses undergo a complex proteolytic maturation on the ER membrane, where the viral RNA is also incorporated into the budding particle. The non-infectious particle is released into the ER lumen and transported via the Golgi complex onto the cellular surface. During transport, the pr domain is cleaved from the prM membrane protein, rendering the particle infectious.

**Figure 2 mps-07-00003-f002:**
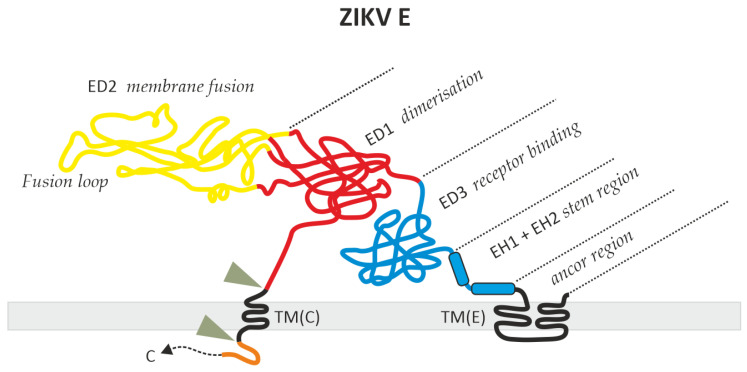
Structural model of the ZIKV-E protein expressed by pME-E2. After organization at the ER membrane, the C-TM(C)-E-TM(E) polyprotein is cleaved by proteases (triangles) and the N-terminus of the E-TM(E) protein is subsequently released. Red, E domain 1 (ED1); Yellow, E domain 2 (ED2); Blue, E domain 3 (ED3) including stem region; Orange, C-terminal parts of the capsid protein; Black, transmembrane (TM) regions. The coding sequence is shown in Figure A1 in the Appendix A section. Dashed arrow, c-terminal part of capsid C.

**Figure 3 mps-07-00003-f003:**
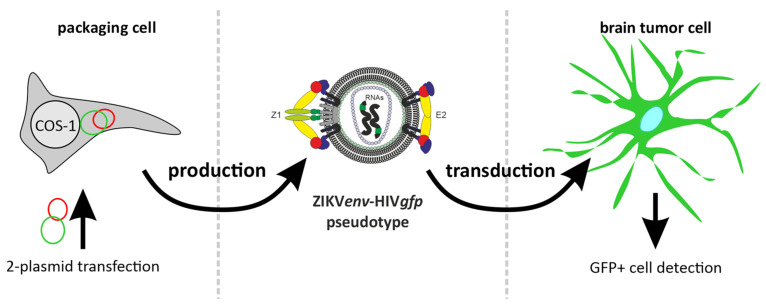
Experimental set-up for the transduction of tumor cells by ZIKVenv-HIVgfp pseudotypes.

**Figure 4 mps-07-00003-f004:**
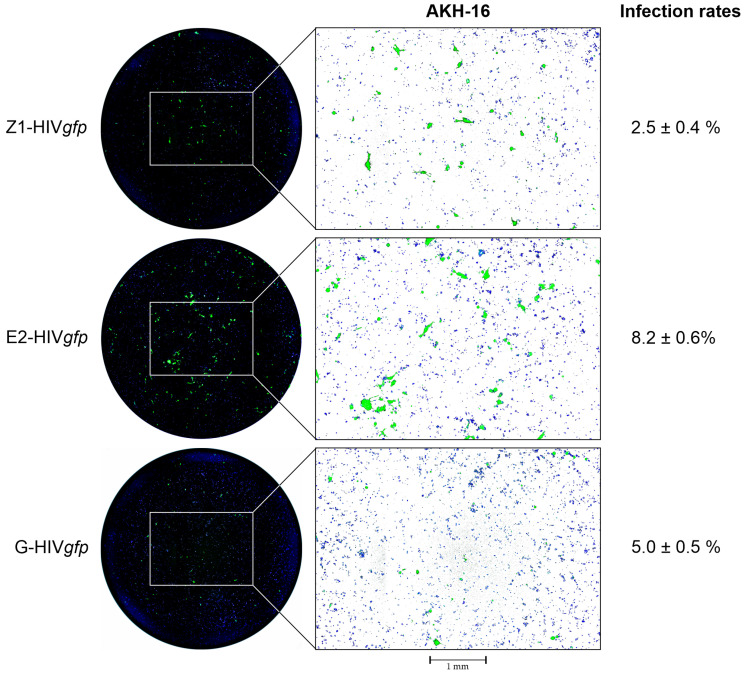
Transduction of the AKH-16 cell culture with HIV*gfp* pseudotypes. Left panel: The scanned and stitched image shows an overview of the GFP+ cells (green) in a 96-well. Cells were inoculated with 100 µL of a cell culture supernatant from pME-env and pNL*gfp*AM transfected COS-1 cells. Pseudotype transduced cells are visible due to their green fluorescence. Cell nuclei were stained blue using 4′,6-diamidino-2-phenylindol (DAPI). Right panel: a section of the 96-well. For better visibility of the blue stained nuclei, the black background was replaced by a white background. Pseudotypes: Z1-HIV*gfp* with envelope prME expressed by pME-Z1; E2-HIV*gfp* with envelope E2 expressed by pME-E2; G-HIV*gfp* with envelope VSV-G expressed by pCMV-VSV-G. Transduction rates (%) were calculated based on five 96-wells.

**Table 1 mps-07-00003-t001:** Transfection mixtures for plasmid DNA and SFA reagents.

Mixture 1 (DNA)		Mixture 2 (SFA)	
Envelope plasmid ^1^	9.25 µL	SFA reagents	15.0 µL
pNL*gfp*AM	2.00 µL	SFA dilution buffer	34.5 µL
SFA dilution buffer	30.00 µL		
	Ʃ 41.25 µL		Ʃ 49.5 µL

^1^ pME-Z1 or pME-E2 or pCMV-VSV-G.

## Data Availability

The authors confirm that the data supporting the findings of this study are all available within the figures and tables of the article.

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
