# Peer review of "Generation of Viral Particles with Brain Cell-Specific Tropism by Pseudotyping HIV-1 with the Zika Virus E Protein"

_mps, 2023, doi:10.3390/mps7010003_

Round 1

Reviewer 1 Report

Comments and Suggestions for Authors

This protocol describes the generation of a ZIKV/HIV-1 pseudo type developed as a new tool for infecting cells derived from the highly malignant brain tumor. This MS is an interesting topic as it is based on the idea that the Zika virus can spreads through the BBB and causes microcephaly. Also, this is about obtaining a higher titer by creating a PV with only E in PrME.

 However, the disappointing thing is that although the protocol was explained very well, it is already known in other PVs such as JEV that PV can be easily made by expressing only E in PrME in Flaviviruses. Please refer other references related to this types of PVs.

In this protocol, human cerebrospinal fluid (hCSF) was used to culture AKH-16 cells. If the reason for using hCSF, which is generally difficult to obtain, is because of the AKH-16 cell culture conditions, the infection rate obtained through it should be a result that specifically reflects the characteristics of the ZIKV E protein. We hope that this special infection test will explain the specificity of brain tumor cells and ZINV/HIV PV.

Additional explanation is needed as to whether this is a receptor characteristic of AKH-16 cells when normalized to the same titer.

In this protocol, human cerebrospinal fluid (hCSF) was used to culture AKH-16 cells. If the reason for using hCSF, which is generally difficult to obtain, is because of the AKH-16 cell culture conditions, the infection rate obtained through it should be a result that specifically reflects the characteristics of the ZIKV E protein. We hope that this special infection test will explain the specificity of brain tumor cells and ZINV/HIV PV.

In other words, an explanation is needed as to whether this is a receptor characteristic of AKH-16 cells when normalized to the same titer in Vero cell.

Author Response

Rev 1              This protocol describes the generation of a ZIKV/HIV-1 pseudo type developed as a new tool for infecting cells derived from the highly malignant brain tumor. This MS is an interesting topic as it is based on the idea that the Zika virus can spreads through the BBB and causes microcephaly. Also, this is about obtaining a higher titer by creating a PV with only E in PrME.

However, the disappointing thing is that although the protocol was explained very well, it is already known in other PVs such as JEV that PV can be easily made by expressing only E in PrME in Flaviviruses. Please refer other references related to this types of PVs.

response:

We have added two references to this topic. In Grunwald et al. 2023 we have constructed ZIKV/JEV chimeras to test if the JEV ED3 domain can enhance infectivity of a PV in the ZIKV background. But JEV and Usutu ED3 were not enhancing infectivity and such PVs were less infective compared to VSV-G. Please see lines 56, 57.

In this protocol, human cerebrospinal fluid (hCSF) was used to culture AKH-16 cells. If the reason for using hCSF, which is generally difficult to obtain, is because of the AKH-16 cell culture conditions, the infection rate obtained through it should be a result that specifically reflects the characteristics of the ZIKV E protein. We hope that this special infection test will explain the specificity of brain tumor cells and ZINV/HIV PV.

response:

hCSF is not that difficult to obtain when you are working together with a department for neurosurgery with the aim to establish primary cell cultures from GBM grad 4 tumors. Some explanation to this topic is added lines 272-76.

Additional explanation is needed as to whether this is a receptor characteristic of AKH-16 cells when normalized to the same titer.

In other words, an explanation is needed as to whether this is a receptor characteristic of AKH-16 cells when normalized to the same titer in Vero cell.

response:

I know this is an important topic. But in this technical protocol we did not want to go into the difficult question of what the ZIKV receptor is in a natural infection. We know that Axl, for example, is one of the proposed receptors, but in animal models it has been shown that Axl is not necessary for ZIKV infection but is present on always all cell lines. It is a phenomenon that occurs between healthy and tumor cells, as all cell lines are more or less cancer cells. Therefore, we think it is important to investigate the infectivity of ZIKV-PVs in healthy brain cells, but this is even more complicated to realize compared to obtaining hCSF or GBM tissue samples.

However, we are always using identical batches of PV supernatants for infection experiments. Aim was to develop PVs that are more active as VSV-g PVs. This was achieved for E PVs. Please see lines 298-305

Reviewer 2 Report

Comments and Suggestions for Authors

Please see the attached file for my comments.

Comments on the Quality of English Language

The overall English is not bad.  Some terms need to be addressed. For example, the HIV Nef is an accessory protein, not an accession protein. 

Author Response

Rev 2              Overall, the description for preparing ZIKV pseudoviruses using a retroviral vector is relatively

clear. Description of the cells used for transfection and the plasmid DNA backbone should be

described in more detail. Generally, the English of the paper needs minor revisions.

response:

The COS and HEK cells used for transfection are standard laboratory cells that are commonly used for the production of pseudotypes. It is not clear to us what special description is required for these standard cells.

We have attached some more data on the AKH-16 cells and the corresponding reference, lines 128-133.

Comments:

Line 251-262: The infection rate seems somewhat low. Can you explain why there is such a

difference?

response:

The aim was to develop a pseudotype for GBM cells that has a higher infection rate than the commonly used VSV-G type. This was achieved. Higher infection rates, to infect more cells can easily be achieved when higher pseudotype concentrations are used but higher PV concentrations are not changing the E/VSV-G ratio. Please see lines 298-305

Line 279: HIV Nef protein is an accessory protein is not accession protein. Please correct.

response:

Sorry, was an oversight (autocorrect)

Line 338-364: In addition to the sequence the plasmid map should be given.

response:

We have added all sequences for Z1 and E2 as well as plasmid maps for pME and pNLgfpAM

Reviewer 3 Report

Comments and Suggestions for Authors

The article, "A method for the preparation of Zika virus E protein encased retroviral particles" by Hai Dang Ngo et al., offers a thorough exploration of the preparation method for Zika virus E protein encased retroviral particles. The introduction adeptly underscores the significance of flaviviruses, with a specific focus on Zika virus . The elucidation of the flavivirus pseudotype concept is particularly commendable, underlining its relevance in virology for diverse applications.

Clarity and Organization: The manuscript exhibits a commendable level of organization and clarity in its presentation. The authors effectively navigate through the subject matter, providing a comprehensive understanding. While the overall flow is well-structured, a minor suggestion is proposed to enhance the coherence of information. Including a concise overview or roadmap of the protocol at the outset of the Methods section would serve as a helpful guide, facilitating a smoother comprehension of the experimental steps for the reader.

Author Response

Rev 3              The article, "A method for the preparation of Zika virus E protein encased retroviral particles" by Hai Dang Ngo et al., offers a thorough exploration of the preparation method for Zika virus E protein encased retroviral particles. The introduction adeptly underscores the significance of flaviviruses, with a specific focus on Zika virus . The elucidation of the flavivirus pseudotype concept is particularly commendable, underlining its relevance in virology for diverse applications.

Clarity and Organization: The manuscript exhibits a commendable level of organization and clarity in its presentation. The authors effectively navigate through the subject matter, providing a comprehensive understanding. While the overall flow is well-structured, a minor suggestion is proposed to enhance the coherence of information. Including a concise overview or roadmap of the protocol at the outset of the Methods section would serve as a helpful guide, facilitating a smoother comprehension of the experimental steps for the reader.

response:

We have added a graphical summary to the section on experimental design. Lines  179-81.

Reviewer 4 Report

Comments and Suggestions for Authors

In this manuscript, the authors described a method for preparing ZIKV E protein pseudotyped HIV-1 and its application in infecting cells derived from highly malignant brain tumor. This method is useful for the research that would employ the ZIKV E protein pseudotyped virus. Below are some questions and suggestions.

1.      In the appendix, the sequence of the pME-E2 open reading frame (ORF) was shown. My suggestion is to show the sequence of the pME-Z1 ORF together in the appendix. If possible, the sequence of the pNLgfpAM plasmid could be uploaded as supplementary materials.

2.      Line 123: “ZIKV can selectively infect and kill glioblastoma cells while leaving normal, healthy brain cells unaffected.” It is better to cite references.

3.      It was widely believed that the prM is critical for rearrangement of ZIKV E proteins into homodimers to facilitate virion maturation and inhibition of the function of prM may result in the loss of ZIKV infectivity. However, the authors found that the pseudotyped virus prepared by pME-E2 which is only encased by ZIKV E protein is still infectious. Could authors give some explanations?

4.      In the last paragraph of the Discussion, the authors told us the ZIKV envelopes are expressed under the control of the CMV promotor with the start sequence (GCC)3ATG in a modified version of pcDNA3.1 which is lacking all the SV40 elements. Have the authors tried another widely used Kozak sequence GCCACCATG? Is it necessary to delete all the SV40 elements in the plasmid for expressing ZIKV envelope by using pcDNA3.1?

Author Response

Rev 4              In this manuscript, the authors described a method for preparing ZIKV E protein pseudotyped HIV-1 and its application in infecting cells derived from highly malignant brain tumor. This method is useful for the research that would employ the ZIKV E protein pseudotyped virus. Below are some questions and suggestions. 

  1. In the appendix, the sequence of the pME-E2 open reading frame (ORF) was shown. My suggestion is to show the sequence of the pME-Z1 ORF together in the appendix. If possible, the sequence of the pNLgfpAM plasmid could be uploaded as supplementary materials. 

response:

We have added all sequences and plasmid maps in the appendix section.

  1. Line 123: “ZIKV can selectively infect and kill glioblastoma cells while leaving normal, healthy brain cells unaffected.” It is better to cite references. 

response:

We have added the reference for this topic. Line 124, 125.

  1. It was widely believed that the prM is critical for rearrangement of ZIKV E proteins into homodimers to facilitate virion maturation and inhibition of the function of prM may result in the loss of ZIKV infectivity. However, the authors found that the pseudotyped virus prepared by pME-E2 which is only encased by ZIKV E protein is still infectious. Could authors give some explanations? 

response:

We have added some explanations to the transmembrane paragraph, lines 298-305. Additionally, we have added the refs for JEV pseudotypes produced with E only, lines 56, 57.

  1. In the last paragraph of the Discussion, the authors told us the ZIKV envelopes are expressed under the control of the CMV promotor with the start sequence (GCC)3ATG in a modified version of pcDNA3.1 which is lacking all the SV40 elements. Have the authors tried another widely used Kozak sequence GCCACCATG? Is it necessary to delete all the SV40 elements in the plasmid for expressing ZIKV envelope by using pcDNA3.1?

response:

We have added the Kozak reference and described the start sequence in more detail in lines 311 and 312.
The difference between GCCGCCgCCATGG and GCCGCCaCCATGG is not that big, as both G and A predominate at this position. We did not test ACC in comparison to GCC. Expression of E and E variants was tested in COS-1 (published in the paper by Grunwald et al. 2023) with high yields.
We have also not yet attempted to increase expression by altering codon usage or codons to improve mRNA transport. This will likely be relevant when the final E construct has been developed that has high activity for tumor cells but preferably no activity for healthy brain cells.

For protein expression it is definitely not necessary to delete SV40 elements. However, we followed a publication by Chang et al 2003, on E protein expression, now also cited as ref 16. This was especially constructed for transient expression (see lines 309, 310. It is of course advantageous to establish cell lines that express E constitutively at the end of all studies, but we are not there yet.

Round 2

Reviewer 1 Report

Comments and Suggestions for Authors

Revision addressed all my comments.

Author Response

thank you

Reviewer 4 Report

Comments and Suggestions for Authors

On the whole, my concerns have been addressed.

Author Response

thank you